

# Functional trait responses to sediment deposition reduce macrofauna-mediated ecosystem functioning in an estuarine mudflat

Sebastiaan Mestdagh[1], Leila Bagaço[1], Ulrike Braeckman[1], Tom Ysebaert[2,3], Bart De Smet[1], Tom
5   Moens[1], Carl Van Colen[1]

[1]Ghent University, Marine Biology Research Group, Krijgslaan 281/S8, B-9000 Ghent, Belgium; [2]Department
of Estuarine and Delta Systems, NIOZ Royal Netherlands Institute for Sea Research and Utrecht University,
P.O. Box 140, 4400 AC Yerseke, The Netherlands;

10   [3]Wageningen University and Research, Wageningen Marine Research, P.O. Box 77, 4400 AB Yerseke, The
Netherlands.

*Correspondence to:* Sebastiaan Mestdagh (sebastiaan.mestdagh@ugent.be)



**Abstract.** Human activities, among which dredging and land use change in river basins, are altering estuarine ecosystems. These activities may result in changes in sedimentary processes, affecting biodiversity of sediment macrofauna. As macrofauna control sediment chemistry and fluxes of energy and matter between water column and sediment, changes in the structure of macrobenthic communities could affect the functioning of an entire ecosystem. We assessed the impact of sediment deposition on intertidal macrobenthic communities and on rates of an important ecosystem function, i.e. sediment community oxygen consumption (SCOC). An experiment was performed with undisturbed sediment samples from the Scheldt river estuary (SW Netherlands). The samples were subjected to four sedimentation regimes: one control and three with a deposited sediment layer of 1, 2 or 5 cm. Oxygen consumption was measured during incubation at ambient temperature. Luminophores applied at the surface, and a seawater-bromide mixture, served as tracers for bioturbation and bioirrigation, respectively. After incubation, the macrofauna was extracted, identified and counted, and classified into functional groups based on motility and sediment reworking capacity. Total macrofaunal densities dropped already under the thinnest deposits. The most affected fauna were surficial and low-motile animals, occurring at high densities in the control. Their mortality resulted in a drop in SCOC, which decreased steadily with increasing deposit thickness, while bioirrigation and bioturbation activity showed increases in the lower sediment deposition regimes, but decreases in the more extreme treatments. The initial increased activity likely counteracted the effects of the drop in low-motile, surficial fauna densities, resulting in a steady rather than sudden fall in oxygen consumption. We conclude that the functional identity in terms of motility and sediment reworking can be crucial in our understanding of the regulation of ecosystem functioning and the impact of habitat alterations such as sediment deposition.

**Key words**: biogeochemical cycling, bioirrigation, bioturbation, ecosystem functioning, functional traits, macrobenthos, SCOC, sediment deposition



## 1 Introduction

It is widely accepted that biodiversity plays an important role in ecosystem functioning. A higher biodiversity can convey a higher resilience and a more efficient functioning of ecosystems in terms of, among others, nutrient cycling and primary productivity (Cardinale et al., 2012; Hooper et al., 2005). Since biodiversity-mediated ecosystem functioning depends on the functional identities of the species present in the community and their densities (Braeckman et al., 2010; Van Colen et al., 2013), functional community descriptors often predict functioning better than taxonomic diversity (Wong and Dowd, 2015). Functional traits, e.g. in terms of motility or sediment reworking rate, can be an indication for a species' behaviour. By being able to rework more or less sediment, species can differentially influence biogeochemical cycling (Wrede et al., 2017). Furthermore, variations in population densities of individual species can influence the ecosystem functioning as well (e.g. Braeckman et al., 2010). Habitat changes that alter densities and/or induce behavioural change of specific functional groups of organisms, e.g. top predators or key players in biogeochemical cycling (Allen and Clarke, 2007; Villnäs et al., 2012), are therefore likely to change the functioning of ecosystems. Natural disturbances occur frequently in coastal and estuarine ecosystems, and recent intense anthropogenic activities often significantly reduce ecosystem resilience (Alestra and Schiel, 2015). An important example of such a human-induced change in coastal and estuarine habitats is sediment deposition. Natural sedimentation is caused by surface runoff from the catchment area or by tidal movements; the former can be intensified by land use change (Thrush et al., 2004). Furthermore, dredging and dumping activities also contribute to sediment deposition, either directly or by creating sediment plumes that subsequently settle down on the seabed (Van Lancker and Baeye, 2015). Such deposition events are expected to alter the productivity of coastal soft-sediment habitats via direct and indirect mechanisms that affect biogeochemical cycling. Firstly, the deposition of fine sediments reduces aerobic mineralisation through the formation of a physical barrier at the sediment-water interface that inhibits re-oxidation of reduced substances in the sediment (Colden and Lipcius, 2015; Hohaia et al., 2014). Macrofauna plays an important role in the biogeochemical cycling of soft sediments through sediment particle mixing (i.e. bioturbation) and the assisted transfer of solutes through the sediment (i.e. bioirrigation) (Braeckman et al., 2010, 2014; Van Colen et al., 2012; Thrush et al., 2006). Both processes can be significantly altered under increased sediment deposition through changes in macrobenthic densities (Alves et al., 2017) or behavior (Rodil et al., 2011). For example, sessile organisms that live attached to the substratum or in tubes, often have a limited capacity to escape burial, and suspension feeders risk clogging of their feeding apparatus (Ellis et al., 2002; Lohrer et al., 2004). Secondly, macrofauna activities can interfere with the deposition induced physical barrier at the sediment-water interface. Sediment deposition induced loss of macrofauna species density and behaviour therefore represents a second, more indirect pathway of how deposition events can alter ecosystem functioning.

The effects of sediment deposition on taxonomic diversity (Thrush et al., 2003), behaviour (Hohaia et al., 2014; Townsend et al., 2014), and ecosystem functioning (Larson and Sundbäck, 2012; Montserrat et al., 2011) have recently received considerable attention. However, to the best of our knowledge, no integrated study of the effect of sediment deposition on the benthic processes that drive biogeochemical cycling (i.e. bioturbation and bioirrigation) has hitherto been published. This study therefore aims to obtain a mechanistic understanding of sediment deposition effects on ecosystem functioning by experimentally assessing the impacts of deposition events of different magnitude (i.e. thickness of the deposited sediment layer) on benthic community diversity and




biological traits (i.e. diversity, densities), benthic processes (i.e. bioturbation and bioirrigation) and biogeochemical cycling in an intertidal soft-sediment habitat. We hypothesize that sediment deposition reduces oxygen availability in the ecosystem underneath, consequently affecting the survival of the macrobenthos and inducing escaping behaviour (Riedel et al., 2008; Villnäs et al., 2012). This may influence biogeochemical cycling,

5    by affecting bioturbation or bioirrigation (Van Colen et al., 2012; Renz and Forster, 2014).

## 2 Materials and Methods

### 2.1 Sample collection and experimental set-up

Samples were collected in March 2015 at the Paulina mudflat (SW Netherlands), which is located along the southern shore of the polyhaline part of the Scheldt estuary (51 ° 21.02 ' N 3 ° 43.78 ' E). The Scheldt estuary experiences a number of human-induced processes that can increase sediment deposition on tidal flats, among which dredging, and the local deposition of dredged sediments at the edges of tidal flats, are some of the most

important examples (De Vriend et al., 2011; van der Wal et al., 2011). The Paulina mudflat harbours a functionally rich benthic macrofaunal community that is numerically dominated by polychaetes (Van Colen et al., 2008).

Twenty-four cylindrical sediment corers (10 cm inner diameter, 29 cm length) were used to randomly collect cores within a 5 x 5 m patch of sediment, consisting of $46 \pm 0.9$ % mud (<63 µm), $22.9 \pm 0.4$ % very fine sand

(63 – 125 µm), $21.7 \pm 0.6$ % fine sand (125 – 250 µm) and $9.4 \pm 0.2$ % medium sand (250 µm – 500 µm). Additional sediment for the experimental deposition treatments had been collected at the same site a few days before the start of the experiment. This additional sediment was sieved over a 1 mm mesh, dried in the lab at 60 °C, heated in a muffle furnace at 500 °C to remove all organic matter (so that treatment effects could be unambiguously assigned to the physical smothering effect), rinsed with demineralized water, and subsequently

sieved again.

All cores were cut to 9 cm, and each core was subsequently subjected to one of four treatments, each with six replicates. Each treatment except the control (T0) consisted of the application of a layer of the pre-treated sediment with a thickness of 1 (T1), 2 (T2) or 5 cm (T5), including a 0.5 cm thick frozen mud cake containing "Magenta" luminophores (Environmental Tracing Systems Ltd., Helensburgh, UK; median grain size 65 µm)

and pre-treated sediment in a 1:1 volume:volume ratio to measure bioturbation activity. The control treatment only received a luminophore cake on top of the natural sediment surface. Seawater from the sampling location (10 °C and a salinity of 20.3, kept still in barrels in the lab for half a day to allow suspended sediment to sink down) was carefully added on top of each core, up to the top edge of the corer. After addition of the water, the added sediment layers compacted to an average of $1.09 \pm 0.18$ (T1), $1.52 \pm 0.10$ (T2) and $3.75 \pm 0.11$ cm (T5),

respectively. The cores were incubated in two tanks under ambient temperature and salinity conditions, filled until half the corer height to buffer for small changes in temperature, and provided with a constant air supply through bubbling underneath the water surface in each core. Each tank had a total capacity of 12 corers, and contained three replicates of each treatment. Oxygen did not penetrate deeper than the lower boundary of the



deposited sediment layers in the deposition treatments, hence the sediment deposition created a physical barrier at the sediment-water interface prohibiting (passive) exchange of dissolved oxygen between the sampled community and the water column at the onset of the experiment. The experiment ran for 15 days, with different measurements taking place during this period. After letting the cores rest to regain biogeochemical equilibria,

sediment oxygen profiles were measured on days 7 and 8, oxygen fluxes on day 12, followed by two days of measuring bioirrigation and a final day on which the cores were sliced for further analysis.

### 2.2 Biogeochemical cycling

For the SCOC measurements, all cores were equipped with a magnetic stirring ring and sealed with an air-tight lid, fitted with two luer stopcocks enabling the sampling of the overlying water for the measurement of sediment-water column exchange of oxygen. During five hours (approximately one-hour intervals), 40-ml water samples were collected through one of the stopcocks using a glass syringe. Replacement water was added by opening the second stopcock and allowing tank water to flow in. The water samples were treated with Winkler reagents

(Parsons et al., 1984) and stored at 4 °C until Winkler titration (Mettler Toledo G20, DGi 101-Mini oxygen electrode, LabX Light Titration software, Columbus, OH, USA). Sediment community oxygen consumption rates (SCOC) were then calculated from the linear decline in oxygen concentration, according to Eq. (1):

$$SCOC = -\frac{dC}{dt}\frac{V}{A} \tag{1}$$

where $\frac{dC}{dt}$ is the change in oxygen concentration in the overlying water (in mmol $L^{-1}$ $d^{-1}$), V is the volume of the

overlying water (in L), and A is the sediment surface area (in $m^2$).

For the measurement of diffusive oxygen uptake (DOU), vertical sediment oxygen profiles were measured with a Unisense OX100 Clark-type needle electrode (Unisense, Aarhus, Denmark). Three profiles were measured in each core and the result was averaged, to account for spatial variability in the sediment. The DOU could then be calculated by multiplying the negative slope of the initial decrease in oxygen concentration, by its diffusion

coefficient (Glud, 2008). The oxygen uptake that could be attributed to macrofaunal respiration was calculated by the formulae described in Mahaut et al. (1995), in which ash-free dry weights (AFDW), calculated from wet weights of the animals (see further) is used to calculate respiration rates:

$$R = 0.0174\ W^{0.0844} \tag{2}$$

where R is the respiration rate in mg C $d^{-1}$ and W the mean individual AFDW in mg C. The amount of carbon was

estimated to be 50 % for all species (Wijsman et al., 1999). Since this formula is only valid for the temperature range of 15 to 20 °C, a $Q_{10}$ of 2 was then assumed to correct the bias, and a respiratory quotient of 0.85 was used to calculate the oxygen consumption, here characterised as faunal uptake (FU; Braeckman et al., 2010; Mahaut et al., 1995). The remaining part of SCOC, after subtraction of DOU and FU, is the biologically-mediated oxygen uptake (BMU), caused indirectly by stimulation of aerobic remineralisation by macrofaunal bioturbation and

irrigation.



### 2.3 Bioirrigation and bioturbation

One day after the oxygen flux measurements, water was siphoned off from each core and replaced by a NaBr-seawater mixture to assess bioirrigation. The NaBr solution had the same density as the seawater; both were mixed

to obtain a solution with a final concentration of 0.1 M NaBr. The solution was added with 100 mL syringes on all cores until as close as possible to the edge, which amounted to 700 ml for T0, T1 and T2, and 600 ml for T5. A first sample of 2 ml was taken immediately after adding the mixture and subsequently after 1, 2, 18 and 21 hours. The bromide concentrations were measured with ion-chromatography and used to calculate bioirrigation rates:

$$Q = -\frac{V_{OW}}{C_{OW} - C_{PW}} \frac{dc_{OW}}{dt} \tag{3}$$

where Q is the bioirrigation rate, $V_{OW}$ is the volume of the overlying water in L, $C_{OW}$ is the initial concentration of bromide in the overlying water (mol L$^{-1}$), $C_{PW}$ the bromide concentration in the pore water and $\frac{dc_{OW}}{dt}$ the change of bromide concentration in the overlying water over time (in mol L$^{-1}$ d$^{-1}$). For $C_{PW}$, an estimation was made by measuring the background concentration in untreated seawater.

On the 14$^{th}$ day of the experiment, the remaining water was siphoned off the cores, which were subsequently sliced per 5 mm from the top until 2 cm into the natural sediment. Deeper slices were cut at a thickness of 10 mm. The sediment in each slice was thoroughly homogenised, after which 5 to 10 mL was sampled and frozen at -20 °C, awaiting further processing for the quantification of bioturbation.

The samples were subsequently dried for 48 hours at 60 °C; water was then carefully added again, after which the

sediment was spread open in a 55 mm inner diameter Petri dish. Each sample was photographed under UV light (365 nm peak wavelength) and luminophores were counted with computer scripts in Matlab v8.1 (MathWorks Inc., 2013) and R (R Development Core Team, 2013). A vertical profile of luminophore pixel counts was constructed for each sediment core and additional R scripts were used to fit the profiles to a non-local bioturbation model from which the biodiffusion coefficient ($D_b^{NL}$, in cm$^2$ d$^{-1}$) was calculated (Wheatcroft et al., 1990). Since

luminophores were only applied on the sediment-water interface, the measured profiles represent disturbance of the surface by bioturbating fauna, rather than providing a total picture of the sediment mixing underneath the surface.

### 2.4 Macrofauna

The remaining 85 to 90 % of the sediment was rinsed over a 500 µm mesh-sized sieve to collect the macrofauna. The animals were stained with a Rose Bengal dye in order to facilitate the identification. Organisms were identified to species level, except for Oligochaeta and *Spio* sp. After identification, all animals were weighed to assess their biomass. The ash-free dry weight (AFDW) was determined by using conversion factors from wet weights

(Sistermans et al., 2006). Biomasses were used to calculate the faunal respiration (Mahaut et al., 1995).





### 2.5 Data analysis

Diversity indices (Shannon-Wiener diversity H' (base e), Pielou's evenness J' and species richness S) were calculated with Primer v6.1 (Clarke and Gorley, 2006). All taxa were assigned to functional groups based on their
motility (from M1 – living fixed in a tube – till M4 – free three-dimensional movement through a burrow system) and sediment reworking activity (surficial modifiers, biodiffusors, upward conveyors and downward conveyors), according to Queiros et al. (2013). All downward conveyors in our study were also classified as upward conveyors, since they can perform both sediment reworking activities

Differences between the treatments for all biotic and abiotic variables, including all species' densities, were first
tested by a 2-way ANOVA, where "Tank" and "Treatment" were used as factors. Since these analyses demonstrated that there were no interaction effects of tank and treatment, a blocked-design ANOVA was applied, with "Tank" as the blocking factor. A Tukey HSD test was used for pairwise comparisons in case of a significant treatment effect. In case the assumptions of normality (tested with a Shapiro-Wilk test) and homogeneity of variances (assessed with Levene's test) for ANOVA were not met, a fourth-root transformation was performed on
the data. Differences in community composition were tested with multivariate two-way permutational analysis of variance (PERMANOVA; Anderson et al. 2008). A Similarity Percentages analysis (SIMPER), based on a Bray-Curtis similarity matrix, was used to determine the species which contributed most to the differences between treatments. When a significant treatment effect was found, pairwise PERMANOVA tests were performed in order to detect differences between the treatments. The PERMANOVA tests were followed by a PERMDISP test to
define whether the found effects are influenced by heterogeneity of multivariate dispersions.

Linear regressions were applied to find relationships between the different response variables. Most importantly, relationships were identified between ecosystem functioning (SCOC), benthic processes (bioturbation, bioirrigation) and the various biotic variables, including densities of all individual species. Further regression tests investigated the contribution of individual species to the density – ecosystem functioning relationship, by using
the densities of all taxa as predictor variables. The optimal model was selected via stepwise combined backward and forward selection. The variance inflation factor (VIF) was used to determine multicollinearity of the predictor variables. All assumptions for linear regression were tested on the residuals and met (no outliers and normal distribution).

All statistical analyses were performed with R v3.0.3 (R Development Core Team, 2013), except the
PERMANOVA and SIMPER tests, for which Primer v6.1 with PERMANOVA+ add-on was used (Clarke and Gorley, 2006).

### 3 Results

### 3.1 Macrofauna

Sediment deposition affected community structure with the community present in T5 differing significantly from the control (2-factor Permanova pseudo-F = 2.457, $P$ = 0.013; pair-wise comparisons T0-5: $P$ = 0.010). The




PERMDISP test was not significant for either the main test or the pair-wise comparison (main test F = 0.858, $P$ = 0.5795; T0-T5: $P$ = 0.6282). Species that contributed most to the dissimilarity in community structure between these treatments were *Aphelochaeta marioni* and Oligochaeta spp. (Table 1). Densities of *Polydora cornuta* (T0: 381.97 ± 131.50 ind m$^{-2}$, T1: 169.77 ± 53.68 ind m$^{-2}$, T2: 42.44 ± 26.84 ind m$^{-2}$, T5: 0 ± 0 ind m$^{-2}$) and *Scrobicularia*

*plana* (T0: 403.19 ± 60.77 ind m$^{-2}$, T1: 381.97 ± 80.53 ind m$^{-2}$, T2: 106.10 ± 51.11 ind m$^{-2}$, T5: 106.10 ± 83.28 ind m$^{-2}$) were significantly lower in T5 (*P. cornuta* T0-T5: $P$ = 0.003, T1-T5: $P$ = 0.014; *S. plana* T0-T5: $P$ = 0.039). The control community had significantly higher total densities than the other communities (T0-T1: $P$ = 0.011, T0-T2: $P$ = 0.043, T0-T5: $P$ = 0.001) while lowest Shannon-Wiener diversity and species richness were found for the T5 community (Fig. 1, Table 2). Community evenness did not differ significantly among treatments.

In general, changes in macrobenthic community composition mirrored differential responses of specific motility and sediment reworking traits (Fig. 2, Table 2). Densities of the two groups of organisms with lowest motility were negatively affected by the applied treatments while densities of more motile species were not significantly different among treatments (Fig. 2a). The density of tube-building organisms (M1) decreased gradually with the thickness of the deposited sediment, whereas densities of species with limited movement (M2) were impaired by

all sediment deposition treatments, irrespective of their magnitude (Fig. 2a).

All sediment reworking groups were affected by the deposition (Fig. 2b). For surficial modifiers, all treatments showed lower densities compared to the control, and for upward conveyors T5 was significantly lower than all other treatments (Surf. Mod. T0-T1: $P$ = 0.033, T0-T2: $P$ = 0.013, T0-T5: $P$ = 0.006; Upw. Conv. T0-T5: $P$ < 0.001, T1-T5: $P$ = 0.009, T2-T5: $P$ = 0.006). The density of biodiffusors was only significantly reduced in T5

compared to the control ($P$ = 0.024)(Fig. 2b).

Activity of the macrofauna (bioturbation and bioirrigation) was significantly affected by the deposition treatments (Table 2). Bioturbation activity was significantly higher in T1 than in all other treatments (T0-T1: $P$ = 0.016, T1-T2: $P$ = 0.048, T1-T5: $P$ = 0.032), and was lowest in T5. While the biodiffusion coefficient $D_b^{NL}$ reached average values in the control treatment, it rose significantly in T1 and dropped again in T2 and T5 (Fig. 3a). A similar

pattern was observed for bioirrigation, but here we only found a significant difference between T1 and T5 ($P$ = 0.019) (Fig. 3b).

### 3.2 Ecosystem functioning

Sediment community oxygen consumption (SCOC) decreased with increasing thickness of the applied sediment layer, ranging from 54.68 ± 5.35 mmol m$^{-2}$ d$^{-1}$ in the control, over 46.79 ± 3.53 mmol m$^{-2}$ d$^{-1}$ in T1 and 44.37 ± 3.52 mmol m$^{-2}$ d$^{-1}$ in T2, to 40.68 ± 3.60 mmol m$^{-2}$ d$^{-1}$ in T5. Only T5 differed significantly from the control ($P$ = 0.030)(Fig. 3c, Table 2). Faunal respiration (FU) accounted for 2.67 ± 1.01 % of the total SCOC in T0, 3.64 ± 1.64 % in T1, 1.75 ± 0.30 % in T2 and 1.99 ± 0.41 % in T5, while the DOU amounted for 18.55 ± 2.64 mmol m$^{-2}$ d$^{-1}$ in

T0, 13.71 ± 1,80 mmol m$^{-2}$ d$^{-1}$ in T1, 11.56 ± 1.79 mmol m$^{-2}$ d$^{-1}$ in T2, and 16.37 ± 1.84 mmol m$^{-2}$ d$^{-1}$ in T5. Neither DOU nor FU showed any significant changes between treatments (Table 2), demonstrating the importance of biotic-mediated oxygen consumption (BMU) in the patterns of total SCOC.





Multiple linear regression showed that the variability in SCOC was significantly related to total macrofaunal density and $D_b^{NL}$, explaining together 54.4% of the variability in SCOC ($P < 0.001$). When total density was divided over the functional groups, we found significant relationships with $D_b^{NL}$ and motility groups M2 and M3 ($P = 0.001$; $R^2 = 0.53$), and with surficial modifiers and biodiffusors ($P < 0.001$; $R^2 = 0.56$). Other variables of

community diversity (Shannon-Wiener diversity, species richness, and Pielou's evenness) were not significant predictors of ecosystem functioning. While no single species was found to contribute significantly to $D_b^{NL}$, a combination of several species contributed significantly to the variability in SCOC ($P < 0.001$; $R^2 = 0.56$). The taxa with a significant contribution were *A. marioni* and *Cyathura carinata* (Table 3). The statistically optimal model for bioirrigation included *Hediste diversicolor* and *P. cornuta* as positive contributors to this process ($P <$

$0.001$; $R^2 = 0.73$)(Table 3).

**4 Discussion**

Tidal flats are dynamic, sedimentary environments that naturally undergo processes of erosion and deposition. Per

tidal cycle, different elevation changes have been observed, e.g. from decreases of 3.3 mm in the Yangtze estuary (China) to increases of 0.6 cm in the estuary of the Seine (France) (Deloffre et al., 2007; Shi et al., 2012). The Scheldt estuary is characterised by its meso- to macro-tidal regime and well-mixed water column. Sediment input from the river basin is relatively low and sand extraction and sea level rise lead to a net export of sediment from the estuary (De Vriend et al., 2011). Sediment accretion on the estuary's tidal flats can amount for about 2 cm yr⁻

¹ (Weerman et al., 2011; Widdows et al., 2004), which suggests that natural sedimentation on the intertidal mudflats is unlikely to exceed even a few millimetres per tidal cycle. More extreme changes in the bed level of mudflats can happen during storm events, either by erosion of the top centimetres of the sediment or by deposition of new sediment (Hu et al., 2015; Marion et al., 2009). Besides natural processes, anthropogenic factors influencing sedimentation are prominent in the estuary, among which dredging in the main channels to ensure access to the

port of Antwerp, and dumping of the dredged material to retain sediment within the estuary, are the most important (Jeuken and Wang, 2010; Meire et al., 2005). Most of this dredged sediment is disposed of near shoals and tidal flats, and can as such affect the intertidal ecosystem (Bolam and Whomersley, 2005; De Vriend et al., 2011; Zheng, 2015). Our results show that even thin sediment deposits can cause a drop in total macrofaunal density, mainly by impacting the highly abundant surface-dwelling animals with low motility (Figs 1-2a,b). These animals, which

belong to reworking and motility class 2 due to their sessile lifestyle (Solan et al., 2004), lack the capacity to escape the deposited sediment and are not adapted to living in deeper sediment layers (Essink, 1999). Since the oxygen penetration depth never exceeded the thickness of the deposited sediment layer, we can assume that oxygen stress was a major driver for the observed decrease in faunal densities. In treatments T1 and T2, oxygen stress was possibly reduced by the increased activity of the macrofauna, due to the animals still being able to disturb the

surface and oxygenate the underlying sediment. Hypoxia can induce escaping behaviour in benthic fauna, as observed in our intermediate treatments, and increase mortality when more severe (Riedel et al., 2008; Villnäs et al., 2012).

Being identified as significant contributors to changes in SCOC, surface-dwelling and low-motile animals are expected to show density patterns similar to those of SCOC itself. However, SCOC only gradually declined with



increasing thickness of the deposited sediment, and this decrease became significant only in the most extreme treatment (T5). Since DOU proved to be constant over all treatments and macrofaunal respiration was negligible compared to the total oxygen consumption, the observed changes in SCOC could be attributed to oxygen uptake caused indirectly by activity of the benthos (i.e. bioturbation and/or bioirrigation). However, both bioirrigation and

bioturbation, the latter of which was linearly related to SCOC, showed that activity increased in treatments T1 and T2. This activity was likely caused by animals for which we found a linear relationship with bioturbation or bioirrigation, like *H. diversicolor*, that are highly mobile and can bury upwards towards the surface, thereby partly irrigating the sediment. *Hediste diversicolor* is a 'gallery-diffusor', which combines biodiffusion in a dense gallery system with biotransport to the bottoms of the tubes (François et al., 2002; Hedman et al., 2011), as well as a well-

known bioirrigator (Kristensen and Hansen, 1999; Riisgaard and Larsen, 2005). Its activity can be expected to result in the oxygenation of deeper sediment layers, but this effect was probably not sufficient to counteract the loss of less mobile, surface-dwelling fauna. Consequently, we observed a gradual and significant decline in SCOC, caused by the disappearance of an abundant group of organisms. Upon addition of the thick sediment layer in treatment T5, species richness dropped significantly and the densities of upward conveyors decreased

considerably, hence preventing the transport of organically rich deep sediment to the surface, through the deposited layer. As a result, the deposited sediment essentially functioned as a barrier, preventing contact between sediment organic matter and oxygen in the water column, and therefore reducing microbial degradation and respiration.

Through alterations in functional trait abundances and community composition, natural and anthropogenic disturbances can affect the entire ecosystem functioning (Bolam et al., 2002; Rodil et al., 2011). In the case of

burial by sediment deposition, our experiment revealed that SCOC can be affected by causing mortality among surface-dwelling and low motile animals, forming the most abundant functional groups of macrobenthos in our system. Macrobenthic diversity and abundance have been shown to exert some control on the magnitude of solute fluxes across the sediment-water interface (Herman et al., 1999; Thrush et al., 2006). Furthermore, previous studies have shown that functional traits of species can be of great importance to explain ecosystem functioning, rather

than or additional to taxonomic diversity (Braeckman et al., 2010; Hooper et al., 2005). Our results highlight the importance of both macrofaunal densities, and the functional identity of species. It is clear that taxonomic diversity alone was not sufficient to explain the changes in ecosystem functioning in our experiment, whereas closer inspection of the functional identities provided more realistic insights.

It should be noted that the sediment we used for deposition was completely defaunated and did not contain organic

matter. Whereas the aim of using defaunated sediment was to allow a better mechanistic understanding of the consequences of sediment deposition, it does not reflect natural conditions. Dredged material from the bottom of the estuary is much richer in organic material and might lead to different results in a similar experiment. Cottrell et al. (2016) showed that benthic species can have a variable tolerance for changes in the enrichment of the sediment, with higher mortalities under high organic loading (and hence likely stronger impacts on macrofauna-

mediated biogeochemical cycling).

**5 Conclusion**





Our experiment revealed new insights into the effects of sediment deposition on the intertidal benthic ecosystem. We found a negative effect on ecosystem functioning, with alterations in macrofauna community structure and activity as the underlying mechanisms. With increasing thickness of the deposited sediment layer, a shift to lower densities of low-motile and surface-dwelling animals resulted in decreased functioning, even though this was

initially dampened by an increased activity of more motile and deeper-living fauna. The latter were responsible for a sustained oxygen penetration through the deposited layer under intermediate treatments, but failed to efficiently do so under more extreme circumstances. It was clear that taxonomic diversity did not suffice to explain changes in functioning, while the functional identity of species did give us important additional insights.

**Data availability**

All data will be deposited in the VLIZ Marine Data Archive (http://mda.vliz.be/introduction.php).

**Author contributions**

SM, LB and CVC devised the experiments. SM and LB carried out the experimental work and collected all data.
SM and CVC led the writing of the manuscript, to which all authors contributed. All authors declare that they do not have any conflict of interest.

**Acknowledgements**

Sebastiaan Mestdagh acknowledges a PhD grant from the Special Research Fund (BOF) of Ghent University. BOF
also provided additional financial support through GOA projects 01GA1911W and 01G02617. Ulrike Braeckman and Carl Van Colen are post-doctoral fellows of the Flemish Research Fund (FWO). We would like to thank the laboratory staff at the Marine Biology Research Group of Ghent University who assisted us with our research: Niels Viaene for help during sampling and Bart Beuselinck for grain size analysis. In addition, we want to thank the lab technicians Jan Sinke and Peter van Breugel at the Royal Dutch Institute for Sea Research (NIOZ –
Yerseke) for analysing bromide samples.

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



TABLE 1: Results from a SIMPER analysis, indicating the contribution of species to the total dissimilarity between treatments. The three species with highest contribution were selected, to achieve a cumulative contribution of over 50 % in all treatment comparisons.

| Treatments | Average dissimilarity | Species | Contribution | Cumulative contribution |
|---|---|---|---|---|
| T0-1 | 42.14 | *Aphelochaeta marioni* | 37.61 % | 37.61 % |
| | | Oligochaeta spp. | 22.36 % | 59.97 % |
| | | *Polydora cornuta* | 5.85 % | 65.83 % |
| T0-2 | 36.49 | *Aphelochaeta marioni* | 37.86 % | 37.86 % |
| | | Oligochaeta spp. | 16.90 % | 54.76 % |
| | | *Polydora cornuta* | 7.24 % | 62.00 % |
| T0-5 | 48.60 | *Aphelochaeta marioni* | 35.25 % | 35.25 % |
| | | Oligochaeta spp. | 22.35 % | 57.60 % |
| | | *Polydora cornuta* | 6.79 % | 64.39 % |
| T1-2 | 38.74 | Oligochaeta spp. | 26.49 % | 26.49 % |
| | | *Aphelochaeta marioni* | 25.53 % | 52.01 % |
| | | *Hediste diversicolor* | 8.02 % | 60.03 % |
| T1-5 | 42.42 | *Aphelochaeta marioni* | 24.20 % | 24.20 % |
| | | Oligochaeta spp. | 21.90 % | 46.10 % |
| | | *Scrobicularia plana* | 10.44 % | 56.55 % |
| T2-5 | 41.15 | Oligochaeta spp. | 31.12 % | 31.12 % |
| | | *Aphelochaeta marioni* | 25.61 % | 56.73 % |
| | | *Hediste diversicolor* | 8.64 % | 65.37 % |



TABLE 2: Results for the factor 'Treatment' from a 2-factor blocked ANOVA tests with 'Treatment' (4 levels) and 'Tank' (2 levels) as factors. M1 till M4 stand for motility classes, as defined by Solan et al., 2004 (M1: living fixed in a tube, M2: sessile, but not fixed in a tube, M3: slow movement through the sediment, M4: free movement in a burrow system). Significant pair-wise differences between treatments are given in the table. In case of heterogeneity of the variances, a fourth root transformation was applied on the data.

| Source | df | SS | MS | F value | P | Pair-wise significance | Transformation |
|---|---|---|---|---|---|---|---|
| M1 | 3 | 74.68 | 24.90 | 12.221 | <0.001* | 0-5, 1-5, 2-5 | Fourth root |
| M2 | 3 | $2.56\times10^7$ | $8.53\times10^6$ | 7.013 | 0.002* | 0-1, 0-2, 0-5 | |
| M3 | 3 | $4.84\times10^6$ | $1.61\times10^6$ | 3.05 | 0.054 | | |
| M4 | 3 | $4.89\times10^5$ | $1.63\times10^5$ | 2.284 | 0.112 | | |
| Surficial modifiers | 3 | $2.00\times10^7$ | $6.68\times10^6$ | 6.087 | 0.004* | 0-1, 0-2, 0-5 | |
| Biodiffusors | 3 | $8.64\times10^6$ | $2.88\times10^6$ | 4.336 | 0.017* | 0-5 | |
| Upward conveyors | 3 | $2.66\times10^6$ | $8.87\times10^5$ | 10.112 | <0.001* | 0-5, 1-5, 2-5 | |
| Downward conveyors | 3 | 86.29 | 28.77 | 24.371 | <0.001* | 0-5, 1-5, 2-5 | Fourth root |
| **Polychaeta** | | | | | | | |
| *Aphelochaeta marioni* | 3 | $1.59\times10^7$ | $5.31\times10^6$ | 4.648 | 0.013* | 0-1, 0-5 | |
| *Eteone longa* | 3 | $1.82\times10^4$ | $6.08\times10^3$ | 1.103 | 0.372 | | |
| *Hediste diversicolor* | 3 | $4.89\times10^5$ | $1.63\times10^5$ | 2.284 | 0.112 | | |
| *Heteromastus filiformis* | 3 | $1.38\times10^5$ | $4.59\times10^4$ | 1.154 | 0.353 | | |
| *Polydora cornuta* | 3 | 53.51 | 17.770 | 7.254 | 0.002* | 0-2, 0-5, 1-5 | Fourth root |
| *Pygospio elegans* | 3 | 44.13 | 14.709 | 5.155 | 0.009* | 0-5, 2-5 | Fourth root |
| *Spio* sp. | 3 | $2.03\times10^3$ | $6.76\times10^2$ | 1 | 0.414 | | |
| *Streblospio benedicti* | 3 | $1.82\times10^4$ | $6.08\times10^3$ | 1.879 | 0.167 | | |
| **Oligochaeta spp.** | 3 | $5.99\times10^6$ | $2.00\times10^6$ | 3.873 | 0.026* | *None* | |
| **Bivalvia** | | | | | | | |
| *Cerastoderma edule* | 3 | $1.08\times10^4$ | $3.60\times10^3$ | 1.583 | 0.226 | | |
| *Limecola balthica* | 3 | $8.84\times10^4$ | $2.95\times10^4$ | 1.939 | 0.158 | | |
| *Scrobicularia plana* | 3 | $4.94\times10^5$ | $1.65\times10^5$ | 5.337 | 0.008* | 0-2, 0-5 | |
| **Gastropoda** | | | | | | | |
| *Peringia ulvae* | 3 | $3.51\times10^4$ | $1.17\times10^4$ | 0.329 | 0.804 | | |
| **Crustacea** | | | | | | | |
| *Bathyporeia pilosa* | 3 | $2.70\times10^3$ | $9.01\times10^2$ | 0.704 | 0.561 | | |
| *Cyathura carinata* | 3 | $1.64\times10^5$ | $5.47\times10^4$ | 1.055 | 0.391 | | |
| $D_b^{NL}$ | 3 | $1.68\times10^{-1}$ | $5.61\times10^{-2}$ | 4.826 | 0.012* | 0-1, 1-2, 1-5 | Fourth root |
| Q | 3 | $3.84\times10^{-3}$ | $1.28\times10^{-3}$ | 4.177 | 0.020* | 1-5 | |
| SCOC | 3 | 632.4 | 210.8 | 3.358 | 0.041* | 0-5 | |
| DOU | 3 | 167.6 | 55.85 | 2.178 | 0.124 | | |
| FU | 3 | 3.50 | 1.17 | 0.869 | 0.475 | | |
| Total density | 3 | $7.18\times10^7$ | $2.39\times10^7$ | 8.346 | 0.001* | 0-1, 0-2, 0-5 | |



| | | | | | | |
|---|---|---|---|---|---|---|
| H' | 3 | $5.01 \times 10^{-1}$ | $1.67 \times 10^{-1}$ | 4.983 | 0.010* | 1-5 |
| J' | 3 | $4.02 \times 10^{-2}$ | $1.34 \times 10^{-2}$ | 2.594 | 0.083 | |
| Species richness | 3 | 36.83 | 12.28 | 6.697 | 0.003* | 0-5, 1-5, 2-5 |

*Significant P-values (P < 0.05) are indicated with ***



TABLE 3: Results of linear regression analyses. SCOC was tested against sets of species (or functional group) densities, and ecosystem processes (bioirrigation and bioturbation). The optimal model was selected via stepwise combined backward and forward selection, based on model AICs. Only significant models ($P$ (slope) < 0.05) were considered.

| Response/predictor | Regression equation | $R^2$ | P |
|---|---|---|---|
| SCOC | | | |
| $x_1$: Total density | $y = 3.35{\times}10^{-3}x_1 + 1.03{\times}10^2x_2 + 25.6$ | 0.544 | 0.0001 |
| $x_2$: $D_b^{NL}$ | | | 0.0224 |
| SCOC | | | |
| $x_1$: M2 | | | 0.0176 |
| $x_2$: M3 | $y = 3.16{\times}10^{-3}x_1 + 5.43{\times}10^{-3}x_2 + 1.02{\times}10^2x_3$ | 0.529 | 0.0404 |
| $x_3$: $D_b^{NL}$ | | | 0.0260 |
| SCOC | | | |
| $x_1$: Surficial modifiers | | | 0.0359 |
| $x_2$: Biodiffusors | $y = 2.92{\times}10^{-3}x_1 + 5.63{\times}10^{-3}x_2 + 1.05{\times}10^2x_3$ | 0.557 | 0.0135 |
| $x_3$: $D_b^{NL}$ | | | 0.0196 |
| SCOC | | | |
| $x_1$: A. marioni | $y = 4.53{\times}10^{-3}x_1 + 2.52{\times}10^{-2}x_2 + 25.9$ | 0.556 | 0.0008 |
| $x_2$: C. carinata | | | 0.0016 |
| Q | | | |
| $x_1$: A. marioni | | | 0.0330 |
| $x_2$: H. diversicolor | | | 0.0002 |
| $x_3$: P. cornuta | $y = -5.76{\times}10^{-6}x_1 + 5.00{\times}10^{-5}x_2 + 3.81{\times}10^{-5}x_3 - 6.33{\times}10^{-5}x_4 - 1.60{\times}10^{-4}x_5 + 2.78{\times}10^{-2}$ | 0.730 | 0.0306 |
| $x_4$: P. elegans | | | 0.0030 |
| $x_5$: S. benedicti | | | 0.0068 |



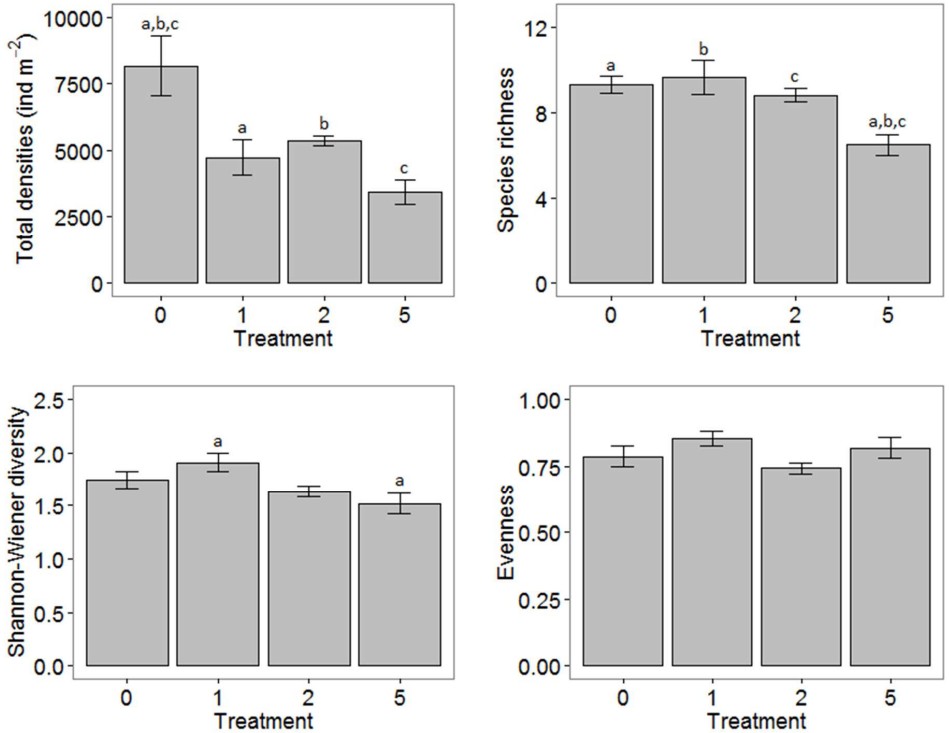

**Figure 1: Bar charts representing total macrofaunal densities (ind m$^{-2}$), species richness, Shannon-Wiener diversity, and Pielou's evenness per treatment. Error bars represent mean ± standard error, letters above the error bars indicate pair-wise significant differences.**





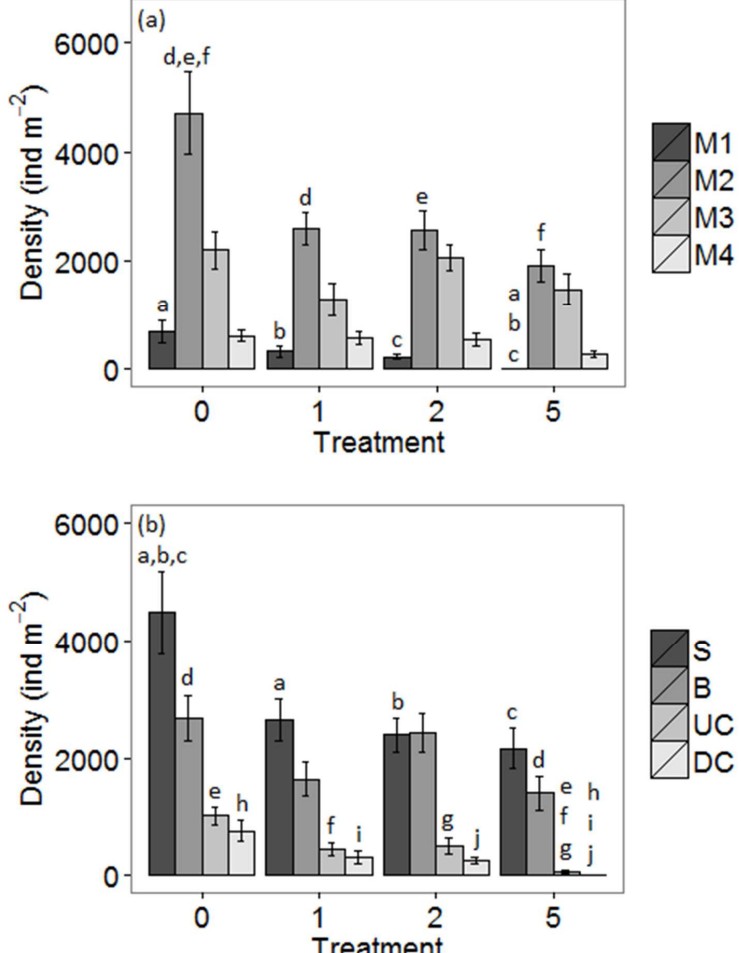

**Figure 2: (a) Bar chart showing the densities of the four motility classes per treatment, in ind m⁻². M1 = organisms living fixed in a tube, M2 = sessile, but not fixed in a tube, M3 = slowly moving organisms, M4 = free movement through a burrow system. (b) Bar chart showing the densities in, ind m⁻², of the four main functional groups, based on sediment reworking activity. S = Surficial modifiers, B = biodiffusors, UC = upward conveyors, DC = downward conveyors. Error bars represent mean ± standard error, letters above the error bars indicate pair-wise significant differences.**

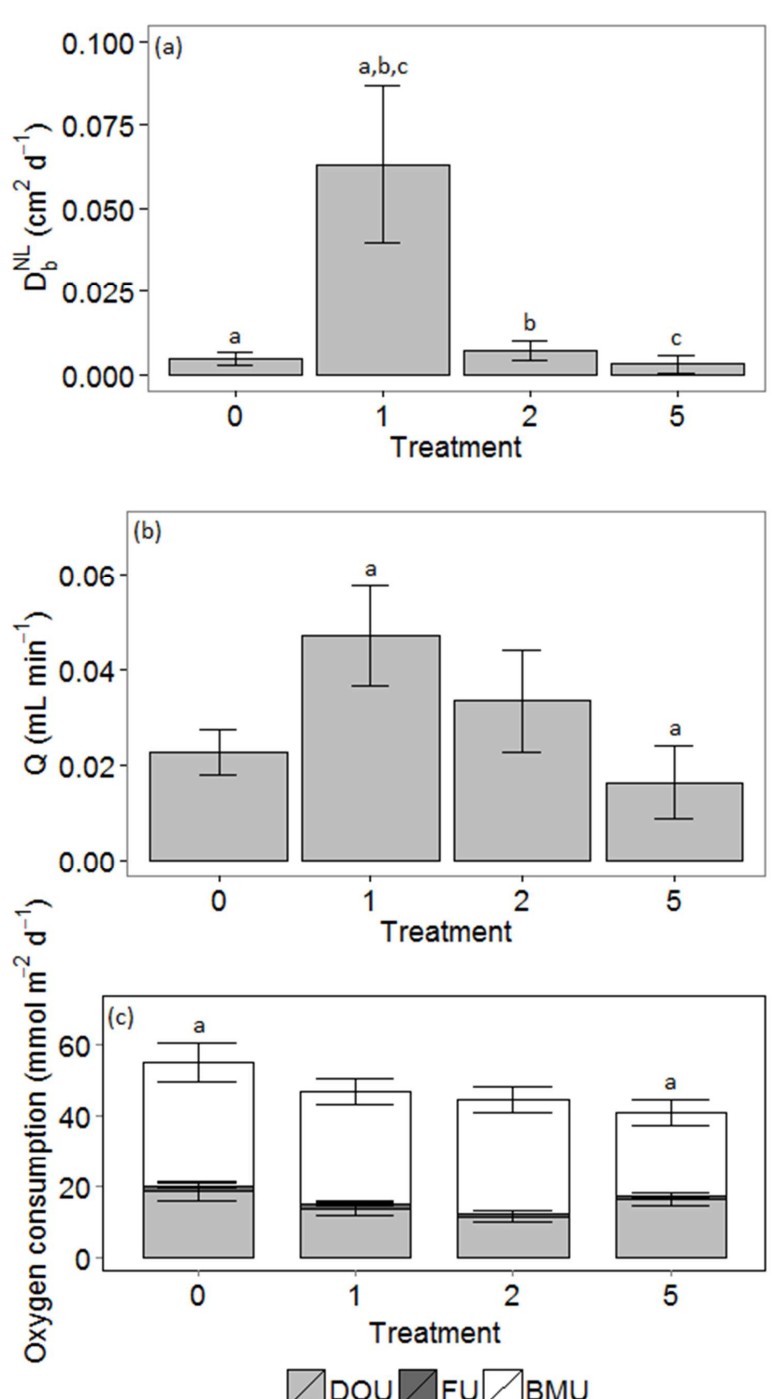



**Figure 3: (a) Bar chart representing the mean bioturbation activity (by means of the biodiffusion coefficient $D_b^{NL}$, in cm² d⁻¹) per treatment ± standard error. (b) Bar chart representing the mean bioirrigation (in mL min⁻¹) per treatment ± standard error. (c) Bar chart representing the mean oxygen consumption (in mmol m⁻² d⁻¹) per treatment ± standard error. The different components of total sediment community oxygen consumption (SCOC) are represented in the chart: diffusive oxygen uptake (DOU), with error bars, faunal uptake (FU), with error bars, and the remaining benthic-mediated oxygen uptake (BMU). The topmost error bars represent the mean ± standard error of the total SCOC (= DOU + FU + BMU). Letters above the error bars indicate pair-wise significant differences.**