# Peer review of "Functional trait responses to sediment deposition reduce macrofauna-mediated ecosystem functioning in an estuarine mudflat"

_Biogeosciences, 2017_

## Referee Comment (RC1) · Anonymous Referee #1 · 29 Nov 2017

Mestdagh and co-workers conducted a laboratory experiment with intact, field-collected cores of intertidal mud to investigate the effects of freshly deposited layers of inert (devoid of organic matter) mud on the structure of the mud's macrofaunal assemblage and the mud's total oxygen consumption. To link the structural with the functional effects of these layers, the authors assessed additional variables: the contributions of species' particle displacement and burrow irrigation to the overall bioturbation of the mud, the oxygen flux across the diffusive boundary layer of the visible mud surface, and the infaunal respiratory oxygen demand. They found that depositing a layer of

inert mud on the surface of mud cores altered the structure of the resident macrofaunal assemblage and so the fauna mediated supply of oxygen to the mud core. They then stress that measures of taxonomic diversity fail to explain the observed functional changes whereas considering species behaviour (functional traits) can lead to mechanistic understanding.

I enjoyed reading this manuscript. Below, please find few requests for clarification.

Page 3, line 22: I understand that a deposit of fine, cohesive sediment will decrease the supply of dissolved oxygen to the deposit-underlying sediment and so decrease the decomposition of organic matter in this sediment with oxygen as electron acceptor. If so, the contribution of anaerobic pathways to the overall decomposition will increase and the upwards diffusing reduced soluble end-products of this decomposition will likely be oxidised with oxygen at the oxic–anoxic boundary somewhere inside the deposit or in the deposit overlying seawater. That is, the re-oxidation of reduced substances (line 24) is not inhibited but simply relocated. Of course, this would not apply for reduced solid phases, but this perhaps needs to be clarified.

Page 4, line 5: In my book, bioturbation includes the displacement of particles and the irrigation of burrows. In line 5, it reads 'bioturbation or bioirrigation', so I assume that the authors do not consider burrow irrigation as a form of bioturbation. Perhaps this needs to be clarified as well.

Page 4, line 27. The authors state that their control (T0) did not receive a layer of pretreated sediment. In line 30, however, they explain that the control did receive a 0.5 cm frozen mud cake, which consisted of pre-treated sediment and luminophores. How did this layer affect the mud–seawater solute exchange and the behaviour of macroinfauna? I feel the authors should discuss this.

Page 5, line 3. The deposit was free of organic matter, so its oxygen demand must have been low increasing the penetration of oxygen into the layer. How do the authors know that this deposit 'prohibited (passive) exchange of dissolved oxygen between

the sampled community and the water column'? Did you measure the penetration of oxygen into the freshly deposited layers with microelectrodes and did you find the oxic–anoxic boundary somewhere inside the layer? If so, how did the four different deposits (0.5, 1, 2, 5 mm) perform in regard to this penetration?

Page 5, line 33. Here, BMU is defined as 'biological-mediated oxygen uptake'. I found this misleading because biological mediated oxygen consumption is also included in estimates of DOU, that is, the consumption by bacterial processes, micro- and meio-fauna. I believe that this contribution to the overall sediment oxygen consumption should be termed 'macrofauna mediated oxygen uptake'.

Page 8. Please consider moving numbers in parentheses to a table; this would improve the readability of your text.

Page 8, line 37. 'biotic-mediated oxygen consumption'. See comment above and please use terms consistently.

Page 9, lines 14–28. I recommend moving this section to the introduction, so the discussion starts with your results.

Page 9, line 31. Please show the oxygen penetration data in the Results section.

Page 23, line 6. 'benthic-mediated oxygen uptake (BMU)'. See comment above and please use terms consistently.

---

## Short Comment (SC1) · 12 Jan 2018

We are grateful for the comments of Anonymous Referee #1 and will take them into account for the revision of our manuscript. All unclear phrasings, definitions and inconsistencies will be changed in the revised version, e.g. the use of the term biologically-mediated oxygen uptake, and what we refer to as bioturbation and bio-irrigation. The referee is mostly concerned about the nature of the added deposits and/or luminophore mud cake and how it influences the underlying community and the mud-seawater exchange of solutes. We will add vertical oxygen profiles to the supplementary material

of the manuscript to clarify the diffusive oxygen consumption in all treatments, including the control that received a 0.5 cm luminophore-spiked deposit. We believe that the presence of this thin deposit in the control – which was needed to compare the same process between treatments and also to avoid experimental artefacts between treatments – did not alter functioning and benthos behaviour in the long term. This conclusion seems justified by quick migration of the benthos through the mud cake and the fact that previous studies in the same habitat and season, that lack a thin deposit, have measured fluxes similar in magnitude to those in our study.

---

## Referee Comment (RC2) · Anonymous Referee #2 · 16 Feb 2018

This is an interesting study. The authors simulated a sudden deposition of sediment and assessed the impact of sediment deposition with different thickness on intertidal microbenthic communities and oxygen consumption. The results showed functional trait could provide important insights on ecosystem functioning, and the taxonomic diversity alone is not sufficient to explain changes. The manuscript is well written and easy to follow. I have only minor concerns about the tables and figures. The tables and figures can be improved. It is expected that your table and figure legends will be quite detailed and very precise. In fact, from the figure title and the axis labels of

a graph/table the reader should be able to determine the question being asked, get a good idea of how the study was done, and be able to interpret the figure without reference to the text. The current table legend is more like discussion, and a mere descriptive of data is preferred. It would be more appropriate that the authors simply provide the objective data, and let the readers to judge whether your conclusion or interpretation are reasonable or not. For example, in Table 1 (page 16) (1) Table legend. The top 3 species with the cumulative contribution (>50%) to the total dissimilarity between treatments. The SIMPER analysis, and the cumulative contribution can be described as footnote. (2) Treatment. Please explain what these treatment mean. For example. What does T0-1 refer to. Please simply state what these data represent. In addition, the cumulative contribution column is redundant and can be deleted, or the contribution column can be deleted because these two column are essentially the same As for Table 2 (page 17) (1) In table legend, Please do not start with "Results….". Please go straight forward what you want to present. For example, statistical factors (2) Is it necessary to show all these factors?

As for Table 3 (page 19). Please consider whether all these equations need to be shown in a table. Maybe the equations could be placed in the supplementary materials

Figure 1. please explain the x axis, i.e., what the treatment of 0, 1, 2 and 5 mean

---

## Author Comment (AC1) · 5 Mar 2018

We wish to thank Anonymous Referee #1 for his thorough and useful comments. We have now revised our manuscript in accordance with the comments raised by the referee. We believe that this revision has substantially improved the quality of the manuscript. Please find below how we have addressed each comment, point by point.

- Referee's comments: 1. Page 3, line 22: I understand that a deposit of fine, cohesive sediment will decrease the supply of dissolved oxygen to the deposit-underlying

sediment and so decrease the decomposition of organic matter in this sediment with oxygen as electron acceptor. If so, the contribution of anaerobic pathways to the overall decomposition will increase and the upwards diffusing reduced soluble end-products of this decomposition will likely be oxidised with oxygen at the oxic–anoxic boundary somewhere inside the deposit or in the deposit overlying seawater. That is, the re-oxidation of reduced substances (line 24) is not inhibited but simply relocated. Of course, this would not apply for reduced solid phases, but this perhaps needs to be clarified.

2. Page 4, line 5: In my book, bioturbation includes the displacement of particles and the irrigation of burrows. In line 5, it reads 'bioturbation or bio-irrigation', so I assume that the authors do not consider burrow irrigation as a form of bioturbation. Perhaps this needs to be clarified as well.

3. Page 4, line 27. The authors state that their control (T0) did not receive a layer of pre-treated sediment. In line 30, however, they explain that the control did receive a 0.5 cm frozen mud cake, which consisted of pre-treated sediment and luminophores. How did this layer affect the mud–seawater solute exchange and the behaviour of macroin-fauna? I feel the authors should discuss this.

4. Page 5, line 3. The deposit was free of organic matter, so its oxygen demand must have been low increasing the penetration of oxygen into the layer. How do the authors know that this deposit 'prohibited (passive) exchange of dissolved oxygen between the sampled community and the water column'? Did you measure the penetration of oxygen into the freshly deposited layers with microelectrodes and did you find the oxic-anoxic boundary somewhere inside the layer? If so, how did the four different deposits (0.5, 1, 2, 5 mm) perform in regard to this penetration?

5. Page 5, line 33. Here, BMU is defined as 'biological-mediated oxygen uptake'. I found this misleading because biological mediated oxygen consumption is also in-cluded in estimates of DOU, that is, the consumption by bacterial processes, microand meio- fauna. I believe that this contribution to the overall sediment oxygen consumption should be termed 'macrofauna mediated oxygen uptake'.

6. Page 8. Please consider moving numbers in parentheses to a table; this would improve the readability of your text.

7. Page 8, line 37. 'biotic-mediated oxygen consumption'. See comment above and please use terms consistently.

8. Page 9, lines 14–28. I recommend moving this section to the introduction, so the discussion starts with your results.

9. Page 9, line 31. Please show the oxygen penetration data in the Results section.

10. Page 23, line 6. 'benthic-mediated oxygen uptake (BMU)'. See comment above and please use terms consistently.

- Authors' response:

1. Indeed, this is correct and we have therefore rephrased this part to clarify better how a physical barrier alters the contribution of anaerobic pathways.

2. Kristensen et al. (2012) proposed to use bioturbation as an umbrella term, incorporating both burrow ventilation and particle reworking. Indeed, burrow ventilation is a mechanism evolved by infauna to enable a constant supply of fresh nutrients and oxygen by pumping overlying water into their burrows, and as a transport process clearly associated with bioturbation. However, since we aimed at disentangling the mechanisms of deposition-induced alteration of SCOC (burrow ventilation, macrofauna respiration or particle mixing into oxic layers) we preferred to distinguish between bioturbation (i.e. particle reworking) and bio-irrigation (i.e. burrow ventilation). We incorporated this rationale in the manuscript.

3. Our objective to disentangle the different mechanisms of altered oxygen consumption necessitated the application of a luminophore-spiked mud cake on all treatments

(including the control sediments). Without such thin cake on the control, the importance of particle mixing and disturbance of the sediment matrix at the sediment-water interface for deposition-altered functioning would have been impossible to investigate. Moreover, though luminophores are in essence inert particles, the absence of such a luminophore mud cake on the natural sediment in the control could potentially have introduced bias between treatments due to species specific responses to e.g. small modifications in physico-chemistry of the sediment matrix, hence creating an experimental artefact. The high survival and appearance of clear bioturbation signs at the sediment surface, already the day after application of the mud cake in the control (photos are included in Supplementary material Annex 2), indicate that the application of the thin deposit evoked fast migration to the sediment-water interface in the control. However, we do not believe that this thin deposition and subsequent fast disturbance related to benthos migration significantly altered functioning at the longer term, i.e. at the end of the experiment 14 days after addition of the mud cakes. This hypothesis is supported by the high survival but lower bioturbation and bio-irrigation in the control as compared to the T1 treatment. Collectively, this suggests a fast recovery of the sediment-water solute exchange following the deposition of the thin mud cake in the control. Indeed the measured oxygen penetration depth and SCOC in the control are comparable in magnitude to the diffusive and sediment community oxygen fluxes measured in the same habitat and season in previous studies (Van Colen et al. 2012; see manuscript for full reference). We have added this rationale in the revised manuscript.

4. The oxygen penetration depth varied from shallower in the control to deeper below the sediment-water interface in the more extreme deposits (that were largely depleted in organic matter as compared to the control). However, oxygen penetration depth remained restricted to the deposited layer for all treatments. Thus, oxygen did not diffuse below the deposited layers into the natural community. The vertical profiles of oxygen penetration are submitted as supplementary material to the manuscript, to which we now refer in the text (Page 5, line 3; Page 9, line 32; See comment 3).

5. We agree that the terminology we used was potentially confusing and have therefore followed the suggestion by this referee to change this term to 'macrofauna-mediated oxygen uptake' in the revised version of the manuscript.

6. We have accepted this comment. We now refer to the new Tables 2 and 4 in the revised manuscript which contain the densities of the macrofauna and the results of the statistical test.

7. This inconsistency apparently remained unnoticed by me and the co-authors, and we have now corrected this throughout the revised manuscript.

8. We have adopted this comment.

9. See also reply to comment 4. Oxygen penetration depths are now provided as supplementary material to the manuscript.

10. Complied with this comment; see also reply to comment 7.

- Authors' changes in the manuscript:

Page 3, line 22: "Firstly, the formation of a physical barrier increases the contribution of anaerobic pathways to the overall decomposition and relocates the re-oxidation of reduced solutes upwards (Colden and Lipcius 2015; Hohaia et al. 2014). Under these circumstances, reduced solid phases would only oxidise when sediment reworking or irrigation of large burrows by macrofauna brings them to the oxic layer."

Page 3, line 27: "Though both processes are interrelated and sometimes grouped under the umbrella term 'bioturbation' (Kristensen et al., 2012), we opted to use them as separate concepts, in order to clearly distinguish between particle reworking and solute transfer. Bioturbation and bio-irrigation can be significantly altered under..."

Page 4, line 31: "... on top of the natural sediment surface. The addition of this mud cake ensured the quantification of particle mixing in these treatments and avoided potential bias between treatments due to species specific responses to the physicochemical environment created by the mud cake. The addition of a luminophore mud cake on top of the sediment surface in the control treatment did not profoundly affect the natural oxygen fluxes or oxygen penetration depth. Our measured values were comparable in magnitude to those of previous studies in the same habitat and season (Van Colen et al. 2012; Annex 1), and clear bioturbation signs on the sediment surface soon after deposition indicate fast migration to the sediment-water interface (Annex 2)."

Please also note the supplement to this comment:
https://www.biogeosciences-discuss.net/bg-2017-417/bg-2017-417-AC1-supplement.pdf

**Supplement:**

ANNEX 1: Vertical oxygen concentration profiles. Depths are shown in μm, oxygen concentrations in μmol L$^{-1}$. All values are means ± standard error. T0 till T5 represent the four different treatments, each with an initial deposited sediment layer of 0 (T0), 1 (T1), 2 (T2) or 5 (T5) cm, that compacted to 1.09 ± 0.18 (T1), 1.52 ± 0.10 (T2) and 3.75 ± 0.11 cm (T5).

| T0 | | T1 | | T2 | | T5 | |
| --- | --- | --- | --- | --- | --- | --- | --- |
| Depth | [O$_2$] | Depth | [O$_2$] | Depth | [O$_2$] | Depth | [O$_2$] |
| -1000 | 267.59 ± 25.01 | -1000 | 260.65 ± 21.94 | -1000 | 269.10 ± 23.18 | -1000 | 276.92 ± 23.56 |
| -750 | 267.49 ± 25.02 | -750 | 260.55 ± 21.92 | -750 | 268.68 ± 23.23 | -750 | 276.78 ± 23.49 |
| -500 | 267.19 ± 24.95 | -500 | 260.56 ± 21.98 | -500 | 268.21 ± 23.55 | -500 | 276.26 ± 23.37 |
| -250 | 266.78 ± 24.82 | -250 | 260.24 ± 22.03 | -250 | 266.67 ± 23.91 | -250 | 275.37 ± 23.17 |
| 0 | 264.96 ± 24.65 | 0 | 258.13 ± 21.47 | 0 | 262.18 ± 22.88 | 0 | 273.68 ± 22.79 |
| 250 | 238.56 ± 25.35 | 250 | 246.07 ± 21.72 | 250 | 249.49 ± 22.78 | 250 | 259.45 ± 24.93 |
| 500 | 198.44 ± 26.64 | 500 | 228.91 ± 23.35 | 500 | 233.66 ± 20.64 | 500 | 234.26 ± 30.29 |
| 750 | 160.76 ± 23.76 | 750 | 196.85 ± 19.06 | 750 | 203.30 ± 16.06 | 750 | 201.92 ± 34.56 |
| 1000 | 129.17 ± 20.30 | 1000 | 159.12 ± 17.56 | 1000 | 177.36 ± 14.29 | 1000 | 173.89 ± 38.78 |
| 1250 | 95.36 ± 15.55 | 1250 | 130.09 ± 18.75 | 1250 | 148.80 ± 16.57 | 1250 | 138.42 ± 31.67 |
| 1500 | 65.00 ± 13.72 | 1500 | 107.64 ± 20.46 | 1500 | 126.25 ± 15.47 | 1500 | 102.02 ± 24.43 |
| 1750 | 42.99 ± 11.08 | 1750 | 91.66 ± 21.93 | 1750 | 107.81 ± 15.90 | 1750 | 80.68 ± 21.82 |
| 2000 | 28.31 ± 8.65 | 2000 | 79.89 ± 22.24 | 2000 | 93.51 ± 16.43 | 2000 | 64.74 ± 20.25 |
| 2250 | 18.41 ± 6.30 | 2250 | 71.11 ± 21.92 | 2250 | 82.51 ± 16.70 | 2250 | 53.01 ± 18.43 |
| 2500 | 11.17 ± 4.31 | 2500 | 63.89 ± 21.51 | 2500 | 73.88 ± 16.02 | 2500 | 45.20 ± 16.37 |
| 2750 | 6.07 ± 2.69 | 2750 | 58.76 ± 20.81 | 2750 | 66.80 ± 15.13 | 2750 | 40.03 ± 14.56 |
| 3000 | 2.45 ± 1.52 | 3000 | 55.00 ± 20.63 | 3000 | 60.45 ± 13.77 | 3000 | 34.16 ± 12.99 |
| 3250 | 1.32 ± 0.79 | 3250 | 51.47 ± 20.11 | 3250 | 55.34 ± 12.25 | 3250 | 29.81 ± 11.56 |
| 3500 | 0.40 ± 0.25 | 3500 | 47.56 ± 19.60 | 3500 | 50.89 ± 11.24 | 3500 | 28.13 ± 10.40 |
| 3750 | 0.00 ± 0.00 | 3750 | 42.90 ± 19.07 | 3750 | 46.39 ± 10.30 | 3750 | 26.48 ± 9.79 |
| 4000 | 0.00 ± 0.00 | 4000 | 39.05 ± 19.08 | 4000 | 42.30 ± 9.16 | 4000 | 25.35 ± 9.65 |
| | | 4250 | 36.54 ± 18.53 | 4250 | 38.54 ± 8.15 | 4250 | 25.58 ± 9.97 |
| | | 4500 | 33.83 ± 17.78 | 4500 | 34.80 ± 7.52 | 4500 | 26.33 ± 10.71 |
| | | 4750 | 31.47 ± 17.00 | 4750 | 31.35 ± 6.82 | 4750 | 27.18 ± 12.00 |
| | | 5000 | 28.82 ± 15.88 | 5000 | 28.93 ± 6.36 | 5000 | 28.28 ± 13.27 |
| | | 5250 | 26.12 ± 14.44 | 5250 | 26.61 ± 5.72 | 5250 | 29.81 ± 14.79 |
| | | 5500 | 23.34 ± 12.92 | 5500 | 24.10 ± 5.22 | 5500 | 31.22 ± 16.62 |
| | | 5750 | 20.00 ± 10.91 | 5750 | 21.67 ± 5.06 | 5750 | 34.40 ± 18.85 |
| | | 6000 | 16.95 ± 9.29 | 6000 | 19.30 ± 5.00 | 6000 | 36.62 ± 19.90 |
| | | 6250 | 17.65 ± 8.07 | 6250 | 17.76 ± 4.88 | 6250 | 38.16 ± 20.66 |
| | | 6500 | 14.21 ± 6.49 | 6500 | 15.63 ± 4.40 | 6500 | 40.22 ± 21.33 |
| | | 6750 | 11.24 ± 5.23 | 6750 | 13.68 ± 3.88 | 6750 | 42.55 ± 22.46 |
| | | 7000 | 8.08 ± 3.91 | 7000 | 12.03 ± 3.68 | 7000 | 45.16 ± 23.84 |
| | | 7250 | 4.98 ± 2.83 | 7250 | 10.28 ± 3.24 | 7250 | 48.35 ± 25.36 |
| | | 7500 | 2.91 ± 2.13 | 7500 | 8.95 ± 3.00 | 7500 | 51.79 ± 26.79 |
| | | 7750 | 1.84 ± 1.82 | 7750 | 8.00 ± 2.97 | 7750 | 54.69 ± 28.15 |
| | | 8000 | 1.30 ± 1.30 | 8000 | 7.10 ± 2.73 | 8000 | 55.57 ± 28.48 |
| | | 8250 | 1.00 ± 1.00 | 8250 | 6.07 ± 2.41 | 8250 | 56.30 ± 28.78 |
| | | 8500 | 0.97 ± 0.97 | 8500 | 5.35 ± 2.14 | 8500 | 57.00 ± 28.91 |
| | | 8750 | 0.10 ± 0.10 | 8750 | 4.79 ± 1.97 | 8750 | 58.51 ± 29.64 |
| | | 9000 | 0.00 ± 0.00 | 9000 | 4.31 ± 1.84 | 9000 | 60.58 ± 30.33 |
| | | 9250 | 0.00 ± 0.00 | 9250 | 3.78 ± 1.80 | 9250 | 61.67 ± 30.73 |
| | | 9500 | 0.00 ± 0.00 | 9500 | 3.07 ± 1.57 | 9500 | 62.61 ± 30.98 |
| | | | | 9750 | 2.93 ± 1.59 | 9750 | 63.37 ± 31.15 |
| | | | | 10000 | 3.08 ± 1.83 | 10000 | 64.44 ± 31.46 |
| | | | | 10250 | 2.72 ± 1.82 | 10250 | 65.09 ± 31.58 |
| | | | | 10500 | 1.96 ± 1.24 | 10500 | 65.71 ± 31.45 |
| | | | | 10750 | 1.91 ± 1.20 | 10750 | 65.98 ± 31.31 |
| | | | | 11000 | 1.64 ± 1.19 | 11000 | 66.26 ± 31.22 |
| | | | | 11250 | 1.21 ± 0.86 | 11250 | 66.41 ± 31.10 |
| | | | | 11500 | 0.25 ± 0.24 | 11500 | 66.57 ± 30.98 |
| | | | | 11750 | 0.17 ± 0.13 | 11750 | 66.81 ± 30.83 |
| | | | | 12000 | 0.24 ± 0.19 | 12000 | 67.08 ± 30.74 |

| | | | |
|---|---|---|---|
| 12250 | $0.11 \pm 0.11$ | 12250 | $67.15 \pm 30.57$ |
| 12500 | $0.04 \pm 0.04$ | 12500 | $67.07 \pm 30.37$ |
| 12750 | $0.00 \pm 0.00$ | 12750 | $66.96 \pm 30.24$ |
| 13000 | $0.00 \pm 0.00$ | 13000 | $66.97 \pm 30.07$ |
| | | 13250 | $66.92 \pm 29.91$ |
| | | 13500 | $66.62 \pm 29.76$ |
| | | 13750 | $66.12 \pm 29.56$ |
| | | 14000 | $65.41 \pm 29.21$ |
| | | 14250 | $64.87 \pm 29.06$ |
| | | 14500 | $64.15 \pm 28.71$ |
| | | 14750 | $63.06 \pm 28.26$ |
| | | 15000 | $61.95 \pm 27.82$ |
| | | 15250 | $60.91 \pm 27.39$ |
| | | 15500 | $59.95 \pm 27.06$ |
| | | 15750 | $59.05 \pm 26.69$ |
| | | 16000 | $58.09 \pm 26.26$ |
| | | 16250 | $57.29 \pm 25.95$ |
| | | 16500 | $56.83 \pm 25.88$ |
| | | 16750 | $56.18 \pm 25.71$ |
| | | 17000 | $54.88 \pm 25.26$ |
| | | 17250 | $53.71 \pm 24.73$ |
| | | 17500 | $52.69 \pm 24.29$ |
| | | 17750 | $51.63 \pm 23.88$ |
| | | 18000 | $50.30 \pm 23.25$ |
| | | 18250 | $49.28 \pm 22.87$ |
| | | 18500 | $48.41 \pm 22.54$ |
| | | 18750 | $47.35 \pm 22.11$ |
| | | 19000 | $46.32 \pm 21.70$ |
| | | 19250 | $45.58 \pm 21.43$ |
| | | 19500 | $44.62 \pm 21.04$ |
| | | 19750 | $43.72 \pm 20.66$ |
| | | 20000 | $42.80 \pm 20.31$ |
| | | 20250 | $41.67 \pm 19.90$ |
| | | 20500 | $40.28 \pm 19.54$ |
| | | 20750 | $39.05 \pm 19.13$ |
| | | 21000 | $38.00 \pm 18.81$ |
| | | 21250 | $37.00 \pm 18.42$ |
| | | 21500 | $35.85 \pm 17.85$ |
| | | 21750 | $34.84 \pm 17.38$ |
| | | 22000 | $33.18 \pm 16.75$ |
| | | 22250 | $32.20 \pm 16.38$ |
| | | 22500 | $30.94 \pm 15.89$ |
| | | 22750 | $29.57 \pm 15.35$ |
| | | 23000 | $28.84 \pm 15.03$ |
| | | 23250 | $27.79 \pm 14.50$ |
| | | 23500 | $27.00 \pm 14.26$ |
| | | 23750 | $26.14 \pm 13.88$ |
| | | 24000 | $25.25 \pm 13.45$ |
| | | 24250 | $24.23 \pm 12.96$ |
| | | 24500 | $23.17 \pm 12.52$ |
| | | 24750 | $21.90 \pm 12.00$ |
| | | 25000 | $20.70 \pm 11.48$ |
| | | 25250 | $19.60 \pm 10.85$ |
| | | 25500 | $18.48 \pm 10.31$ |
| | | 25750 | $17.53 \pm 9.88$ |
| | | 26000 | $16.65 \pm 9.44$ |
| | | 26250 | $15.76 \pm 8.98$ |
| | | 26500 | $15.25 \pm 8.71$ |
| | | 26750 | $14.58 \pm 8.33$ |
| | | 27000 | $13.98 \pm 8.02$ |

| | |
|---|---|
| 27250 | $13.19 \pm 7.60$ |
| 27500 | $12.43 \pm 7.19$ |
| 27750 | $11.67 \pm 6.75$ |
| 28000 | $11.00 \pm 6.35$ |
| 28250 | $10.15 \pm 5.87$ |
| 28500 | $9.20 \pm 5.34$ |
| 28750 | $8.52 \pm 4.93$ |
| 29000 | $8.02 \pm 4.68$ |
| 29250 | $7.47 \pm 4.38$ |
| 29500 | $7.07 \pm 4.17$ |
| 29750 | $6.74 \pm 4.00$ |
| 30000 | $6.19 \pm 3.68$ |
| 30250 | $5.84 \pm 3.44$ |
| 30500 | $5.56 \pm 3.29$ |
| 30750 | $5.21 \pm 3.10$ |
| 31000 | $4.93 \pm 2.95$ |
| 31250 | $4.29 \pm 2.63$ |
| 31500 | $3.54 \pm 2.20$ |
| 31750 | $3.21 \pm 1.97$ |
| 32000 | $3.12 \pm 1.91$ |
| 32250 | $2.55 \pm 1.60$ |
| 32500 | $2.31 \pm 1.44$ |
| 32750 | $1.97 \pm 1.23$ |
| 33000 | $1.61 \pm 1.00$ |
| 33250 | $1.23 \pm 0.76$ |
| 33500 | $0.89 \pm 0.55$ |
| 33750 | $0.43 \pm 0.29$ |
| 34000 | $0.17 \pm 0.17$ |
| 34250 | $0.07 \pm 0.07$ |
| 34500 | $0.00 \pm 0.00$ |

ANNEX 2: Photos from the surface of the control cores, taken the first, second, and seventh day of the experiment. The six rows, labeled 0.1 till 0.6, represent the six replicates of the control treatment.

[Figure]

| 0.6 |
[Figure]
 |

---

## Author Comment (AC2) · 5 Mar 2018

We are thankful for the useful comments of Anonymous Referee #2. We have revised the figure and table captions accordingly.

- Referee's comments: The current table legend is more like discussion, and a mere descriptive of data is preferred. It would be more appropriate that the authors simply provide the objective data, and let the readers to judge whether your conclusion or interpretation are reasonable or not.

1. For example, in Table 1 (page 16) (1) Table legend. The top 3 species with the cumulative contribution (>50%) to the total dissimilarity between treatments. The SIMPER analysis, and the cumulative contribution can be described as footnote. (2) Treatment. Please explain what these treatment mean. For example. What does T0-1 refer to. Please simply state what these data represent. In addition, the cumulative contribution column is redundant and can be deleted, or the contribution column can be deleted because these two column are essentially the same.

2. As for Table 2 (page 17) (1) In table legend, Please do not start with "Results ...". Please go straight forward what you want to present. For example, statistical factors (2) Is it necessary to show all these factors?

3. As for Table 3 (page 19). Please consider whether all these equations need to be shown in a table. Maybe the equations could be placed in the supplementary materials.

4. Figure 1. please explain the x axis, i.e., what the treatment of 0, 1, 2 and 5 mean.

- Authors' response:

1. We deleted the 'Contribution' column from the table and rewrote the caption according to the referee's suggestions. In addition, we added a footnote to the table.

2. We agree that a mere representation of F- and p-values could be sufficient for a decent understanding of our results. All unnecessary columns (i.e. df, SS and MS) were therefore deleted in the revised manuscript. We suggest the following caption, based on the referee's comment.

3. We opted to present the results of the linear regressions entirely in the main manuscript, as it provides a complete understanding of the strength and direction of the relationship between the response and predictor variables. Therefore, we would prefer to keep the results as they are currently presented in the table. In accordance to Comment 2, we opted to rewrite the caption of this table.

4. We added this extra information to the figure caption of this and the next two figures.

- Authors' changes in the manuscript:

Caption Table 1: "The three species with highest cumulative contribution (> 50 %) to the total dissimilarity between treatments. The first column shows the treatments being compared (e.g. T0-1: a comparison between treatments T0 and T1)."

Footnote Table 1: "Results from a SIMPER analysis."

Caption Table 2: "Statistical factors from 2-factor blocked ANOVA tests with 'Treatment' (4 levels) and 'Tank' as factors. M1 till M4 stand for motility classes, as defined by Solan et al. (2004) (M1: living fixed in a tube, M2: sessile, but not fixed in a tube, M3: slow movement through the sediment, M4: free movement in a burrow system). Significant pair-wise differences between treatments are given in the table. All results for species and functional groups are given for densities."

Caption Table 3: "Linear regressions of sediment community oxygen consumption (SCOC) against sets of species (or functional group) densities, and ecosystem processes (bio-irrigation - Q - and bioturbation - D_b^NL), and of bio-irrigation against the densities of species. Only significant models (P (slope) < 0.05) were considered. M2 and M3 are motility classes as defined by Solan et al. (2004) – M2: sessile, but not fixed in a tube, M3: slow movement through the sediment."

Captions Figures 1-3: "... The four treatments represent the thickness of the applied sediment layer (in cm)."

---

## Author Response (AR1)

We wish to thank Anonymous Referee #1 for his thorough and useful comments. We have now revised our manuscript in accordance with the comments raised by the referee. We believe that this revision has substantially improved the quality of the manuscript. Please find below how we have addressed each comment, point by point.

**Comment 1**    *Page 3, line 22: I understand that a deposit of fine, cohesive sediment will decrease the supply of dissolved oxygen to the deposit-underlying sediment and so decrease the decomposition of organic matter in this sediment with oxygen as electron acceptor. If so, the contribution of anaerobic pathways to the overall decomposition will increase and the upwards diffusing reduced soluble end-products of this decomposition will likely be oxidised with oxygen at the oxic–anoxic boundary somewhere inside the deposit or in the deposit overlying seawater. That is, the re-oxidation of reduced substances (line 24) is not inhibited but simply relocated. Of course, this would not apply for reduced solid phases, but this perhaps needs to be clarified.*

Indeed, this is correct and we have therefore rephrased this part to clarify better how a physical barrier alters the contribution of anaerobic pathways:

Page 3, line 22: "*Firstly, the formation of a physical barrier increases the contribution of anaerobic pathways to the overall decomposition and relocates the re-oxidation of reduced solutes upwards (Colden and Lipcius 2015; Hohaia et al. 2014). Under these circumstances, reduced solid phases would only oxidise when sediment reworking or irrigation of large burrows by macrofauna brings them to the oxic layer.*"

**Comment 2**    *Page 4, line 5: In my book, bioturbation includes the displacement of particles and the irrigation of burrows. In line 5, it reads 'bioturbation or bio-irrigation', so I assume that the authors do not consider burrow irrigation as a form of bioturbation. Perhaps this needs to be clarified as well.*

Kristensen et al. (2012) proposed to use bioturbation as an umbrella term, incorporating both burrow ventilation and particle reworking. Indeed, burrow ventilation is a mechanism evolved by infauna to enable a constant supply of fresh nutrients and oxygen by pumping overlying water into their burrows, and as a transport process clearly associated with bioturbation. However, since we aimed at disentangling the mechanisms of deposition-induced alteration of SCOC (burrow ventilation, macrofauna respiration or particle mixing into oxic layers) we preferred to distinguish between bioturbation (i.e. particle reworking) and bio-irrigation (i.e. burrow ventilation). We incorporated this rationale in the manuscript:

Page 3, line 27: "*Though both processes are interrelated and sometimes grouped under the umbrella term 'bioturbation' (Kristensen et al., 2012), we opted to use them as separate concepts, in order to clearly distinguish between particle reworking and solute transfer. Bioturbation and bio-irrigation can be significantly altered under…*"

**Comment 3**    *Page 4, line 27. The authors state that their control (T0) did not receive a layer of pre-treated sediment. In line 30, however, they explain that the control did receive a 0.5 cm frozen mud cake, which consisted of pre-treated sediment and luminophores.*

*How did this layer affect the mud–seawater solute exchange and the behaviour of macroinfauna? I feel the authors should discuss this.*

Our objective to disentangle the different mechanisms of altered oxygen consumption necessitated the application of a luminophore-spiked mud cake on all treatments (including the control sediments). Without such thin cake on the control the importance of particle mixing and disturbance of the sediment matrix at the sediment-water interface for deposition-altered functioning would have been impossible to investigate. Moreover, though luminophores are in essence inert particles, the absence of such a luminophore mud cake on the natural sediment in the control could potentially have introduced bias between treatments due to species specific responses to e.g. small modifications in physico-chemistry of the sediment matrix, hence creating an experimental artefact. The high survival and appearance of clear bioturbation signs at the sediment surface, already the day after application of the mud cake in the control (photos are included in Supplementary material Annex 2), indicate that the application of the thin deposit evoked fast migration to the sediment-water interface in the control. However, we do not believe that this thin deposition and subsequent fast disturbance related to benthos migration significantly altered functioning at the longer term, i.e. at the end of the experiment 14 days after addition of the mud cakes. This hypothesis is supported by the high survival but lower bioturbation and bio-irrigation in the control as compared to the T1 treatment. Collectively, this suggests a fast recovery of the sediment-water solute exchange following the deposition of the thin mud cake in the control. Indeed the measured oxygen penetration depth and SCOC in the control are comparable in magnitude to the diffusive and sediment community oxygen fluxes measured in the same habitat and season in previous studies (Van Colen et al. 2012; see manuscript for full reference). We have added this rationale in the revised manuscript:

Page 4, line 31: *"… on top of the natural sediment surface. The addition of this mud cake ensured the quantification of particle mixing in these treatments and avoided potential bias between treatments due to species specific responses to the physico-chemical environment created by the mud cake. The addition of a luminophore mud cake on top of the sediment surface in the control treatment did not profoundly affect the natural oxygen fluxes or oxygen penetration depth. Our measured values were comparable in magnitude to those of previous studies in the same habitat and season (Van Colen et al. 2012; Annex 1), and clear bioturbation signs on the sediment surface soon after deposition indicate fast migration to the sediment-water interface (Annex 2)."*

**Comment 4** *Page 5, line 3. The deposit was free of organic matter, so its oxygen demand must have been low increasing the penetration of oxygen into the layer. How do the authors know that this deposit 'prohibited (passive) exchange of dissolved oxygen between the sampled community and the water column'? Did you measure the penetration of oxygen into the freshly deposited layers with microelectrodes and did you find the oxic-anoxic boundary somewhere inside the layer? If so, how did the four different deposits (0.5, 1, 2, 5 mm) perform in regard to this penetration?*

The oxygen penetration depth varied from shallower in the control to deeper below the sediment-water interface in the more extreme deposits (that were largely depleted in organic matter as compared to the control). However, oxygen penetration

depth remained restricted to the deposited layer for all treatments. Thus, oxygen did not diffuse below the deposited layers into the natural community. The vertical profiles of oxygen penetration are submitted as supplementary material to the manuscript, to which we now refer in the text (Page 5, line 3; Page 9, line 32; See comment 3).

**Comment 5**  *Page 5, line 33. Here, BMU is defined as 'biological-mediated oxygen uptake'. I found this misleading because biological mediated oxygen consumption is also included in estimates of DOU, that is, the consumption by bacterial processes, micro- and meio-fauna. I believe that this contribution to the overall sediment oxygen consumption should be termed 'macrofauna mediated oxygen uptake'.*

We agree that the terminology we used was potentially confusing and have therefore followed the suggestion by this referee to change this term to 'macrofauna-mediated oxygen uptake' in the revised version of the manuscript.

**Comment 6**  *Page 8. Please consider moving numbers in parentheses to a table; this would improve the readability of your text.*

We have accepted this comment. We now refer to Tables 2 and 4 in the revised manuscript which contain the results of the statistical test.

**Comment 7**  *Page 8, line 37. 'biotic-mediated oxygen consumption'. See comment above and please use terms consistently.*

This inconsistency apparently remained unnoticed by me and the co-authors, and we have now corrected this throughout the revised manuscript.

**Comment 8**  *Page 9, lines 14–28. I recommend moving this section to the introduction, so the discussion starts with your results.*

We have adopted this comment.

**Comment 9**  *Page 9, line 31. Please show the oxygen penetration data in the Results section.*

See also reply to comment 4. Oxygen penetration depths are now provided as supplementary material to the manuscript.

**Comment 10**  *Page 23, line 6. 'benthic-mediated oxygen uptake (BMU)'. See comment above and please use terms consistently.*

Complied with this comment; see also reply to comment 7.

We are thankful for the useful comments of Anonymous Referee #2. We have revised the figure and table captions accordingly:

***The current table legend is more like discussion, and a mere descriptive of data is preferred. It would be more appropriate that the authors simply provide the objective data, and let the readers to judge whether your conclusion or interpretation are reasonable or not.***

**Comment 1** *For example, in Table 1 (page 16) (1) Table legend. The top 3 species with the cumulative contribution (>50%) to the total dissimilarity between treatments. The SIMPER analysis, and the cumulative contribution can be described as footnote. (2) Treatment. Please explain what these treatment mean. For example. What does T0-1 refer to. Please simply state what these data represent. In addition, the cumulative contribution column is redundant and can be deleted, or the contribution column can be deleted because these two column are essentially the same.*

We deleted the 'Contribution' column from the table and rewrote the caption as follows, according to the referee's suggestions:

*"The three species with highest cumulative contribution (> 50 %) to the total dissimilarity between treatments. The first column shows the treatments being compared (e.g. T0-1: a comparison between treatments T0 and T1)."*

We added a footnote to the table:

*"Results from a SIMPER analysis."*

**Comment 2** *As for Table 2 (page 17) (1) In table legend, Please do not start with "Results ...". Please go straight forward what you want to present. For example, statistical factors (2) Is it necessary to show all these factors?*

We agree that a mere representation of F- and p-values could be sufficient for a decent understanding of our results. All unnecessary columns (i.e. df, SS and MS) were therefore deleted in the revised manuscript. We suggest the following caption, based on the referee's comment:

*"Statistical factors from 2-factor blocked ANOVA tests with 'Treatment' (4 levels) and 'Tank' as factors. M1 till M4 stand for motility classes, as defined by Solan et al. (2004) (M1: living fixed in a tube, M2: sessile, but not fixed in a tube, M3: slow movement through the sediment, M4: free movement in a burrow system). Significant pair-wise differences between treatments are given in the table. All results for species and functional groups are given for densities."*

**Comment 3** *As for Table 3 (page 19). Please consider whether all these equations need to be shown in a table. Maybe the equations could be placed in the supplementary materials.*

We opted to present the results of the linear regressions entirely in the main manuscript, as it provides a complete understanding of the strength and direction of the relationship between the response and predictor variables. Therefore, we would prefer to keep the results as they are currently presented in the table.

In accordance to Comment 2, we opted to rewrite the caption of this table:

*"Linear regressions of sediment community oxygen consumption (SCOC) against sets of species (or functional group) densities, and ecosystem processes (bio-irrigation - Q - and bioturbation - $D_b^{NL}$), and of bio-irrigation against the densities of species. Only significant models (P (slope) < 0.05) were considered. M2 and M3 are motility classes as defined by Solan et al. (2004) – M2: sessile, but not fixed in a tube, M3: slow movement through the sediment."*

**Comment 4**     *Figure 1. please explain the x axis, i.e., what the treatment of 0, 1, 2 and 5 mean.*

We added the next sentence to the figure caption of this and the next two figures:

[revised manuscript text omitted]
 = -5.76\times10^{-6}x_1 + 5.00\times10^{-5}x_2 + 3.81\times10^{-5}x_3 - 6.33\times10^{-5}x_4 - 1.60\times10^{-4}x_5 + 2.78\times10^{-2}$ | 0.730 | 0.0306 |
| $x_4$: P. elegans | | | 0.0030 |
| $x_5$: S. benedicti | | | 0.0068 |

[Figure]

**Figure 1: Bar charts representing total macrofaunal densities (ind m⁻²), species richness, Shannon-Wiener diversity, and Pielou's evenness per treatment. Error bars represent mean ± standard error, letters above the error bars indicate pair-wise significant differences. The four treatments represent the thickness of the applied sediment layer (in cm).**

[Figure]

**Figure 2: (a) Bar chart showing the densities of the four motility classes per treatment, in ind m$^{-2}$. M1: organisms living fixed in a tube, M2: sessile, but not fixed in a tube, M3: slowly moving organisms, M4: free movement through a burrow system. (b) Bar chart showing the densities in, ind m$^{-2}$, of the four main functional groups, based on sediment reworking activity. S: Surficial modifiers, B: biodiffusors, UC: upward conveyors, DC: downward conveyors. Error bars represent mean ± standard error, letters above the error bars indicate pair-wise significant differences. The four treatments represent the thickness of the applied sediment layer (in cm).**

[Figure]

**Figure 3: (a) Bar chart representing the mean bioturbation activity (by means of the biodiffusion coefficient $D_b^{NL}$, in cm$^2$ d$^{-1}$) per treatment ± standard error. (b) Bar chart representing the mean bio-irrigation (in mL min$^{-1}$) per treatment ± standard error. (c) Bar chart representing the mean oxygen consumption (in mmol m$^{-2}$ d$^{-1}$) per treatment ± standard error. The different components of total sediment community oxygen consumption (SCOC) are represented in the chart: diffusive oxygen uptake (DOU), with error bars, faunal uptake (FU), with error bars, and the remaining macrofauna-mediated oxygen uptake (MMU). The topmost error bars represent the mean ± standard error of the total SCOC (= DOU + FU + BMU). Letters above the error bars indicate pair-wise significant differences. The four treatments represent the thickness of the applied sediment layer (in cm).**